# Seminal Plasma, Sperm Concentration, and Sperm-PMN Interaction in the Donkey: An In Vitro Model to Study Endometrial Inflammation at Post-Insemination

**DOI:** 10.3390/ijms21103478

**Published:** 2020-05-14

**Authors:** Jordi Miró, Henar Marín, Jaime Catalán, Marion Papas, Sabrina Gacem, Marc Yeste

**Affiliations:** 1Equine Reproduction Service, Department of Animal Medicine and Surgery, Faculty of Veterinary Sciences, Autonomous University of Barcelona, E-08193 Bellaterra (Cerdanyola del Vallès), Spain; henarmarin@hotmail.com (H.M.); dr.jcatalan@gmail.com (J.C.); papas.marion@gmail.com (M.P.); swp.sabrina.gacem@gmail.com (S.G.); 2Biotechnology of Animal and Human Reproduction (TechnoSperm), Institute of Food and Agricultural Technology, University of Girona, E-17003 Girona, Spain; marc.yeste@udg.edu; 3Unit of Cell Biology, Department of Biology, Faculty of Sciences, University of Girona, E-17003 Girona, Spain

**Keywords:** seminal plasma, sperm, polymorphonuclear neutrophils, donkey

## Abstract

In the donkey, artificial insemination (AI) with frozen-thawed semen is associated with low fertility rates, which could be partially augmented through adding seminal plasma (SP) and increasing sperm concentration. On the other hand, post-AI endometrial inflammation in the jenny is significantly higher than in the mare. While previous studies analyzed this response through recovering Polymorphonuclear Neutrophils (PMN) from uterine washings, successive lavages can detrimentally impact the endometrium, leading to fertility issues. For this reason, the first set of experiments in this work intended to set an in vitro model through harvesting PMN from the peripheral blood of jennies. Thereafter, how PMN, which require a triggering agent like formyl-methionyl-leucyl-phenylalanine (FMLP) to be activated, are affected by donkey semen was interrogated. Finally, we tested how four concentrations of spermatozoa (100 × 10^6^, 200 × 10^6^, 500 × 10^6^ and 1000 × 10^6^ spermatozoa/mL) affected their interaction with PMN. We observed that semen, which consists of sperm and SP, is able to activate PMN. Whereas there was a reduced percentage of spermatozoa phagocytosed by PMN, most remained attached on the PMN surface or into a surrounding halo. Spermatozoa not attached to PMN were viable, and most of those bound to PMN were also viable and showed high tail beating. Finally, only sperm concentrations higher than 500 × 10^6^ spermatozoa/mL showed free sperm cells after 3 h of incubation, and percentages of spermatozoa not attached to PMN were higher at 3 h than at 1 h, exhibiting high motility. We can thus conclude that semen activates PMN in the donkey, and that the percentage of spermatozoa phagocytosed by PMN is low. Furthermore, because percentages of spermatozoa not attached to PMN were higher after 3 h than after 1 h of incubation, we suggest that PMN-sperm interaction plays an instrumental role in the reproductive strategy of the donkey.

## 1. Introduction

Most European donkey breeds are currently in danger of extinction. However, there has been, in the recent years, an increased global demand for donkey products. As such, donkey skin, which is used in China to make a traditional donkey-hide gelatin (Ejiao); meat; and milk, which represents an alternative to its bovine counterpart for infants with some allergic and inflammatory conditions, are also attracting the interest of the cosmetic industry. Despite this, our current knowledge of the donkey reproductive physiology is still scarce and, despite horse reproduction technologies being used, many differences between these two species exist.

Jackasses show high spermatogenic efficiency [1], and their sperm concentration, total cell number, sperm kinematic parameters and percentages of progressively motile and morphologically normal spermatozoa are higher than in stallions [2,3]. In addition, while frozen-thawed donkey spermatozoa have good post-thaw viability and motility [4,5,6], are able to penetrate zona pellucida free bovine oocytes in vitro [7] and yield good conception rates (50%) following artificial insemination (AI) of mares [1], their pregnancy rates are poor (0–28%) when used to inseminate jennies [8].

Oliveira et al. investigated the most suitable jackass sperm concentration for inseminating jennies and mares and concluded that increasing insemination doses up to 10^9^ spermatozoa/mL augmented fertility rates in jennies (four inseminations per cycle), achieving figures similar to those obtained in mares [9]. In addition, these authors demonstrated that in order for optimal fertility of frozen-thawed donkey sperm to be reached, deep-horn intrauterine insemination should be used together with four inseminations per cycle (concentration: 10^9^ spermatozoa/mL).

Natural breeding and AI induce a physiological endometrial inflammatory response, entailing neutrophils influx into the uterus. Kotilainen et al. observed that infusing spermatozoa rather than bacteria into the mare uterus provoked the influx of Polymorphonuclear neutrophils (PMN) [10]. They also reported that the highest PMN infiltration occurred when mares were inseminated with frozen-thawed or concentrated fresh semen, and that the intensity of the PMN reaction relied on sperm concentration and/or insemination volume. Therefore, the deposited semen is responsible for the uterine inflammatory response, triggering strong chemotaxis of PMN along with cytokine expression soon after insemination [11,12,13,14,15]. Related with this, it is worth mentioning that when inseminated with frozen-thawed semen, jennies show an exacerbated inflammatory response, higher than in mares [16], which was suggested to lead to reduced pregnancy rates [17].

On the other hand, not only does removing seminal plasma (SP) result in an improvement of quality and survival of cooled-stored donkey semen [18], but is also a required step prior to sperm cryopreservation [4]. However, SP has positive effects on sperm motility and survival into the uterus, since it helps avoid oxidative damage [19] and decreases the binding of sperm to PMN and other phagocytic cells [5,20,21,22]. Despite this, the mechanism through which SP modulates endometrial inflammation remains to be elucidated. Moreover, and as explained before, sperm concentration and velocity are higher in the donkey than in the horse, and their relationship with the interaction between PMN and spermatozoa is also unknown. In fact, our current knowledge comes from in vitro studies focused on sperm:PMN binding in the donkey, the rich PMN samples being obtained through uterine washings at 6 h post-AI [22,23]. However, performing too many uterine washings may produce significant injuries on the endometrium and reduce the reproductive performance, which is especially critical in the case of endangered breeds.

Against this background, the objectives of the present study were: (1) to set a method aimed at obtaining a PMN- rich fraction from jenny peripheral blood; and (2) to determine whether the whole semen, with the presence of SP, and different sperm concentrations affect the percentages of spermatozoa attached to PMN.

## 2. Results

### 2.1. Experiment 1: Recovering Donkey PMN from Peripheral Blood Samples

We tested seven protocols, as described in Materials and Methods. Protocols 1 to 4 yielded concentrations lower than 100 × 10^3^ PMN/µL. Similarly, protocols 5 and 6 also harvested concentrations lower than 100 × 10^3^ PMN/µL, although they were slightly higher in some biological replicates. In contrast, Protocol 7 showed significantly (*p* < 0.05) higher PMN concentrations than the others, with values greater than 100 × 10^3^ cells/µL in all cases (Table 1). 

### 2.2. Experiment 2: Evaluation of PMN-sperm Binding (FMLP and DMSO)

#### 2.2.1. Sperm Motility

Sperm motility was evaluated through Computer-Assisted Sperm Analysis (CASA). However, we found difficulties while using this device, since CASA systems track the head to determine sperm movement, and a significant number of sperm cells were attached to PMN and showed tail beating. Therefore, it was required for us to distinguish between attached and non-attached (free) sperm populations.

Evaluation of sperm motility in the free population, which was evaluated by CASA, evidenced that after 3 h and 4 h of incubation, most spermatozoa of this population were immotile. However, a high percentage of spermatozoa attached to PMN showed high tail beating and, occasionally, some were released and exhibited high velocity and progressiveness (average path velocity, VAP ≥ 95 µm/s; percentage of straightness, STR ≥ 73%; percentage of linearity, LIN ≥ 65%; amplitude of lateral head displacement, ALH ≤ 2.7 µm; and frequency of head displacement, BCF ≥ 13.5 Hz). Interestingly, only treatments with sperm concentrations higher than 500 × 10^6^ sperm/mL showed free motile spermatozoa after 3 h of incubation (Appendix A). For this reason, the relationship between PMN and spermatozoa was investigated in treatments containing 500 × 10^6^ sperm/mL (Table 2; Figure 1, Figure 2, Figure 3, Figure 4 and Figure 5).

#### 2.2.2. Sperm Viability

Sperm viability decreased significantly (*p* < 0.05) after 3 h of incubation, irrespective of the treatment (Table 2). In addition, no significant differences (*p* > 0.05) in the viability of free spermatozoa, were observed between treatments. On the other hand, dimethyl sulfoxide (DMSO) alone or in the presence of PMN showed no detrimental effects on sperm survival.

At the beginning of the experiment (0 h), percentages of viable spermatozoa attached to PMN in the presence of FMLP were significantly (*p* < 0.05) higher than in the other treatments. In addition, although percentages of viable spermatozoa bound to PMN in the treatment containing DMSO were significantly (*p* < 0.05) lower at 1 h and higher at 3 h than in the other ones, no significant differences between treatments were observed at 2 h and 4 h (Figure 1). In the case of the treatment containing SP, neither variations along the incubation time, nor significant differences with other treatments after 2 h and 4 h of incubation were observed (*p* > 0.05; Figure 1).

#### 2.2.3. Sperm-PMN Binding

Percentages of spermatozoa attached to PMN increased along the incubation time, reaching significantly (*p* < 0.05) higher values with regard to the beginning of the experiment after 4 h of incubation. However, the presence of FMLP or DMSO did not differ from the treatment that did not contain these two agents (i.e., PMN+SP; Figure 2). Moreover, the number of spermatozoa attached to PMN (sperm:PMN ratio) showed a progressive decrease along the incubation period, and was significantly (*p* < 0.05) higher in the PMN + SP treatment than in the ones containing FMLP or DMSO after 0 h, 2 h and 3 h of incubation (Figure 3).

#### 2.2.4. Sperm Attachment to PMN via Their Head or Tail

Percentages of spermatozoa attached to PMN via sperm head varied between 37% and 52% on average (Figure 4). At 0 h, percentages of spermatozoa attached to PMN via sperm head were significantly (*p* < 0.05) higher in the treatment containing SP than in that containing SP and DMSO. In contrast, at 1 h, no significant differences (*p* > 0.05) in the percentages of spermatozoa attached to PMN via sperm head were observed between treatments. Whereas those percentages were significantly (*p* < 0.05) higher in the treatment with SP than in that containing SP and FMLP after 2 h of incubation, these differences were not observed at 3 h, when percentages of spermatozoa attached to PMN via sperm head were significantly (*p* < 0.05) higher in the treatment containing SP and DMSO than in those containing either SP, or SP, and FMLP. Finally, at the end of the experiment, data resembled to those of 0 h, since percentages of spermatozoa attached to PMN via sperm head were significantly (*p* < 0.05) higher in the treatment containing SP than in that containing SP and DMSO (Figure 4).

Percentages of spermatozoa attached to PMN via sperm tail ranged between 46% and 69% on average, and significant (*p* < 0.05) differences between incubation times and between treatments within the same incubation time were found (Figure 5). In effect, percentages of spermatozoa bound to PMN via the sperm tail at 0 h were significantly (*p* < 0.05) higher in the treatment with SP (PMN + SP) than in that containing SP and DMSO (PMN + SP + DMSO). In addition, whereas percentages of spermatozoa bound to PMN via the tail were significantly (*p* < 0.05) higher after 1 h and 3 h than at 0 h, no significant different differences (*p* > 0.05) between 0 h, 2 h and 4 h were observed. After 4 h of incubation, the treatment containing DMSO showed significantly (*p* < 0.05) higher percentages of spermatozoa bound to PMN via the tail than the other two treatments (Figure 5).

### 2.3. Experiment 3: Evaluation of PMN-sperm Binding (Sperm Concentration)

#### Sperm-PMN Binding

Since results from Experiment 2 showed no significant effects from the presence of FMLP or DMSO, the evaluation of how different sperm concentrations (100 × 10^6^, 250 × 10^6^, 500 × 10^6^, and 1000 × 10^6^ spermatozoa/mL) affected sperm-PMN binding only took into consideration the treatments that did not contain either FMLP or DMSO.

Percentages of spermatozoa attached to PMN were significantly (*p* < 0.05) higher in the treatment containing 1000 × 10^6^ spermatozoa/mL than in those containing lower sperm concentrations, at 0 h and 1 h. However, these differences between sperm concentrations were not observed after 2 h of incubation. Regardless of sperm concentration, sperm:PMN ratios decreased along the incubation time, reaching the lowest levels after 3 h of incubation (Figure 6).

With regard to sperm viability, percentages of viable spermatozoa ranged from 30% to 50% on average, without much variation throughout the incubation time (Figure 7). While samples with the highest sperm concentration (1000 × 10^6^ spermatozoa/mL) showed the highest percentage of viable spermatozoa attached to PMN at 0 h and 1 h, no significant differences with the other concentrations were observed after 2 h of incubation.

## 3. Discussion

Endometrial inflammation in the jenny after insemination was previously investigated in vitro using uterine flushing collected at 6 h post-AI [22]. Although an enriched PMN fraction from that fluid was obtained (100 × 10^3^ cells/mL on average) in the aforementioned study, continued uterine washings can detrimentally impact the female reproductive tract and, in turn, cause infertility. Since the Catalan donkey is an endangered breed and in order to avoid the aforementioned undesirable side effect, the first aim of the current work was to set an in vitro model by isolating PMN from the peripheral blood. From the seven protocols tested, one (which was designated as Protocol 7) clearly showed the highest recovery PMN rate, yielding ≥ 100 × 10^3^ cells/mL. As distinct from our results, other protocols tested in other species [24,25], including the equine [26,27], were proven to be unable to isolate enough PMN from the peripheral blood.

In vitro experiments evaluating the activity of PMN in the horse previously used different activating agents, such as formyl-methionyl-leucyl-phenylalanine (FMLP) or recombinant equine interleukin 8 (reqIL-8) [26]. In the current study, while the whole semen (including spermatozoa and SP) was found to be able to stimulate PMN, the presence of FMLP did not alter the percentages of spermatozoa bound to PMN, nor was the viability of free and bound-to-PMN sperm populations over the incubation time affected. These data are in line with previous reports conducted in donkeys, in which SP was found to modulate sperm-PMN binding and sperm motility [22]. On the other hand, the decrease in sperm survival throughout the incubation time was independent from the presence of PMN. While the viability of the sperm population attached to PMN at 0 h was higher using FMLP, perhaps as a result of a fast PMN stimulation, no differences between treatments and over incubation were observed. This suggests that the viability of the sperm population attached to PMN is higher than that of the unattached (free) sperm population.

Remarkably, we observed that few spermatozoa appeared to be phagocytosed by PMN, since while many attached the PMN surface, only some sperm cells were found to be within a stained halo surrounding less colored PMN. While one could not completely exclude that there was non-observed phagocytosis, which would need further specific studies including time-lapse microscopy, it was very apparent that an important proportion of spermatozoa remained surrounding PMN. Polymorphonuclear neutrophils are the first barrier in front of pathogens, via phagocytosis or through releasing their own DNA, histones, and enzymes, including cathepsin (CAT), elastase (ELA) and myeloperoxidase (MPO). In the latter case, this creates complexes known as neutrophil extracellular traps (NETs; NETosis), which prevent the dissemination of pathogens [28,29,30,31,32]. According to Branzk et al. [33], whether PMN trigger phagocytosis or NETosis depends on the size of the pathogen; in effect, while the contact of bacteria with PMN induces phagocytosis, the presence of larger pathogenic cells (such as the yeast ones) activates NETosis. The sperm:PMN binding observed herein warrants further research on two different realms. On the one hand, studies on the potential ligand/receptor mechanism through which PMN triggers phagocytosis upon sperm:PMN binding are required to confirm our reduced percentages of sperm phagocytosis by PMN. On the other hand, and because the formation of NETs was reported on in humans and other species [20,34,35,36,37], further research on whether and how NETs are formed when PMN and sperm are incubated in the donkey is much warranted. NETs are known to trap mammalian spermatozoa in the vicinity of PMNs (i.e., without direct contact), which results in a significant loss of sperm motility [34]. For this reason, we suggest that in a similar fashion to that investigated in other species [31,32,38], additional studies focused upon staining of DNA and PMN enzymes should be conducted. Remarkably, addressing this issue is of high relevance in this context, since the dynamics of sperm:PMN binding observed in our study suggests that sperm phagocytosis could be inhibited, which, as indicated by Branzk et al. [33], would lead to the activation of NETosis. Indeed, deficiencies in the activity of phagocytic receptors were suggested to disrupt the size-dependent selectivity of NET-release due to inefficient phagocytosis. Moreover, blocking of Dectin-1, a phagocytic receptor present in human PMN, with a specific antibody is known to lead to lower rates of phagocytosis and higher NETosis response. Based on the low percentages of phagocytosis, we suggest that seminal plasma factors could inhibit PMN receptors involved in the induction of phagocytosis, which would lead to the activation of NETosis in a similar fashion to that reported by Branzk et al. [33] when Dectin-1 is inhibited.

In the present study, we observed that a significant number of spermatozoa attached to PMN were viable and showed tail beating. In addition, not only were the percentages of viable spermatozoa higher in the bound-to-PMN than in the free sperm population, but sperm cells that were apparently released from PMN after 3 h of incubation also showed good motility. Moreover, we also noticed that during the first hours of incubation, spermatozoa tended to attach PMN through the tail rather than through the head. Since we do not know whether this observation has any relationship with the sperm ability to release from PMN, future studies should also contemplate, through time-lapse microscopy, the binding dynamics between spermatozoa and PMN and the relevance of each sperm compartment. All these data suggest that the attachment of spermatozoa to PMN does not always result in the clearance of the formers, but this apparent transient binding could play an important role in the donkey reproduction strategy, since it is quite likely that sperm cells released from PMN are able to fertilize.

At the beginning of the experiment, percentages of spermatozoa bound to PMN were significantly higher in the treatment containing the highest sperm concentration (1000 × 10^6^ spermatozoa/mL). This could be explained by the higher likelihood of a given PMN to interact with a given spermatozoon, as sperm concentrations increase. However, it is worth noting that these differences in sperm:PMN binding observed between sperm concentrations were not found after 3 h of incubation. In addition, whereas both sperm:PMN ratios and percentages of viable spermatozoa decreased throughout incubation, percentages of spermatozoa attached to PMN increased over successive incubation times. These data indicate that PMN could be progressively activated throughout incubation, thereby attaching a higher number of spermatozoa. Nevertheless, percentages of viable spermatozoa attached to PMN (approximately 50%) showed no variations between sperm concentrations and incubation times. This suggests that the higher the sperm concentration, the higher the possibility of observing free spermatozoa after 3 h and 4 h of incubation. This hypothesis would be in agreement with our CASA observations, as only concentrations higher than 500 × 10^6^ sperm/mL showed free motile spermatozoa at the end of the experiment.

Oliveira et al. observed an increase in fertility rates after inseminating jennies with high concentrated sperm doses [9]. Since jackasses usually show good semen quality with higher sperm concentration and motility than in stallions [1,3], one could suggest that this better sperm quality could play an important physiologic role for the post-breeding endometrial reaction in the jenny. Indeed, as explained before, high sperm concentrations with good motility appear to be required in order for spermatozoa to be released from PMN after binding. However, an important question herein is whether spermatozoa that ultimately fertilize the oocyte belong to the sperm population that are released from PMN. Therefore, future studies should also focus on the relationship between PMN and spermatozoa at post-breeding in the donkey.

## 4. Materials and Methods

### 4.1. Animals

Three Catalonian jackasses, aged 5–11 years, and three Catalonian jennies, aged 5–8 years, were used in this study. All animals were in good health conditions and of proven fertility. While conducting experiments, they were housed at the experimental farm, Faculty of Veterinary Science, Autonomous University of Barcelona (Bellaterra, Cerdanyola del Vallès; Spain). Jackasses were maintained in individual paddocks, whereas jennies were grouped in a big paddock. Animals were fed grain forage, straw and hay, and water was provided ad libitum. The study was approved by the Ethics Committee, Autonomous University of Barcelona (Bellaterra, Cerdanyola del Vallès, Spain; Code: CEEAH 1424, 23.04.2018).

### 4.2. Experiment 1. Isolation of Donkey PMN from Peripheral Blood

Peripheral blood was collected from three different jennies via jugular venipuncture with 10 mL BD Vacutainer^®^ tubes containing 18.0 mg of Ethylenediaminetetraaceticacid (EDTA; BD-Plymouth, UK). Eight tubes were collected per jenny and protocol, and seven protocols were tested. Samples were protected from the light, and immediately transported to the laboratory at room temperature within the following 10 min. To isolate PMN and adjust the concentration to 100 × 10^3^ PMN/mL, different protocols used in other species were evaluated. Each protocol was replicated seven times.

#### 4.2.1. Protocol 1

This protocol was based on the method set by Baumber et al. [26] to isolate equine neutrophils. Briefly, whole blood samples were centrifuged at 200× *g* and room temperature for 15 min. Supernatants (blood plasma) were discarded and pellets containing leukocytes and erythrocytes (2.5 mL) were layered onto four tubes containing 2.5 mL of 59% and 2.5 mL of 75% Percoll solutions (Sigma-Aldrich^®^, Merck KGaA, Darmstadt, Germany). Samples were centrifuged at 400× *g* and room temperature for 20 min. Neutrophils were collected from the interface of the 59/75% Percoll gradient, resuspended in 5 mL PBS and centrifuged at 200× *g* and room temperature for 10 min. Pellets were resuspended in 1 mL PBS and samples were analyzed through a hematological cytometer (ADVIA 120 Siemens Medical Solutions; Fernwald, Germany) to determine the concentration of PMN per mL.

#### 4.2.2. Protocol 2

This protocol was based on the method set by Siemsen et al. [25] designed to isolate non-human neutrophils. Blood samples were collected in tubes containing EDTA and then incubated at 37 °C in a water bath for 30 min, which was the time required for red blood cells to stack. The leukocyte-rich plasma fraction was aspirated and layered onto four tubes containing 2.5 mL of 70% and 2.5 mL of 85% Percoll solutions. Samples were centrifuged at 400× *g* and room temperature for 20 min. Neutrophils were collected from the interface of the 70/85% Percoll gradient, resuspended in 2 mL PBS and washed twice at 200× *g* for 10 min with PBS. The resulting pellets were resuspended in 1 mL PBS and then analyzed through a hematological cytometer (ADVIA 120 Haemathology System, Siemens Healthcare GmbH, Erlangen, Germany) to determine the concentration of PMN per mL.

#### 4.2.3. Protocol 3

The third protocol was based on that of Roth and Kaeberle, which was set to isolate bovine PMN [24]. In brief, blood samples collected with vacutainer tubes were gathered in 15-mL conical tubes and centrifuged at 400× *g* for 15 min. The leukocyte layer was aspirated with some plasma and placed onto 4 mL of Ficoll (Hystopaque 1077^®^, Sigma-Aldrich^®^, Merck KGaA, Darmstadt, Germany); samples were centrifuged at 300× *g* and room temperature for 30 min. Supernatants were discarded and the white layer just below the Ficoll one was aspirated and transferred into a clean tube. Samples were added with two volumes of distilled water, mixed and incubated at room temperature for 5–7 s to lyse the residual erythrocytes. Following this, samples were immediately mixed in an equal volume of 1.7% NaCl solution to restore isotonicity, and subsequently centrifuged at 250× *g* for 10 min. Supernatants were discarded and pellets were resuspended in 1 mL of Lactate Ringer (B Braun, Rubí, Barcelona, Spain). Resuspended pellets were analyzed using a hematological cytometer (ADVIA 120 Siemens Medical Solutions) to determine the concentration of PMN per mL.

#### 4.2.4. Protocol 4

This protocol was based on that from Loftus et al. [27], which aims at isolating equine PMN. Blood samples were cooled down and maintained at 4 °C until visibly sedimented, which lasted for approximately 20 min. The leukocyte rich plasma layer was transferred into a new tube and centrifuged at 250× *g* and 4 ºC for 10 min. Thereafter, the pellet was resuspended in 40 mL of ice-cold 0.9% NaCl, and the resulting cell suspension was layered onto 10 mL Hystopaque 1077^®^ (Sigma- Aldrich^®^) and centrifuged at 300× *g* and room temperature for 40 min. The liquid layer was discarded and the pellet was resuspended in 4 mL ice-cold PBS. Following this, 1 mL of the resulting cell suspension was layered onto 9 mL of 90% Percoll and centrifuged at 650× *g* and room temperature for 40 min. The neutrophil fraction was collected, mixed with 4 mL PBS, and centrifuged at 300× *g* for 8 min. Pellets were subsequently resuspended in 1 mL PBS, and then analyzed with a hematological cytometer (ADVIA 120 Siemens Medical Solutions) to determine the concentration of PMN per mL.

#### 4.2.5. Protocol 5

The fifth protocol was a modification from Protocol 1 which as previously mentioned (see Section 4.2.1), was based on Loftus et al. [27]. In brief, blood samples were collected in tubes containing EDTA. Samples were incubated in a water bath at 37 °C for 30 min to allow red blood cells to stack, forming rouleaux. The leukocyte-rich plasma layer was subsequently aspirated, transferred into 15-mL conical tubes, and centrifuged at 402× *g* and 4 °C for 5 min. Cells were resuspended with 4 mL ice-cold PBS and then centrifuged at 250× *g* and 4 °C for 5 min. Resulting pellets were resuspended in 500 µL ice-cold PBS containing 2% paraformaldehyde, and maintained on ice for 1 h. Thereafter, cells were centrifuged and resuspended in 500 µL PBS, and the resulting sample was analyzed with a hematological cytometer (ADVIA 120 Siemens Medical Solutions) to determine the concentration of PMN per mL, prior to adjusting the concentration to 100 × 10^6^ PMN/mL in ice-cold PBS.

#### 4.2.6. Protocol 6

This protocol consisted of a modification of Protocol 2, which was based on that of Loftus et al. [27] for isolating equine PMN (see Section 4.2.2). With this purpose, blood samples were collected in tubes containing EDTA and then incubated in water bath at 37 °C for 30 min to allow red blood cells to stack. The leukocyte-rich plasma layer was aspirated, transferred into 15-mL conical tubes and centrifuged at 402× *g* and 4 °C for 5 min. Samples were resuspended in 4 mL ice-cold PBS and then centrifuged at 250× *g* and 4 °C for 5 min. Thereafter, cells were resuspended in 500 µL ice-cold PBS, and then analyzed with a hematological cytometer (ADVIA 120 Siemens Medical Solutions) to determine the concentration of PMN per mL. Cell concentration was adjusted to 100 × 10^6^ PMN/mL in ice-cold PBS.

#### 4.2.7. Protocol 7

Blood samples were collected in tubes containing EDTA and incubated in a water bath at 37 °C for 30 min, to stack red blood cells. The leukocyte-rich plasma layer was aspirated, transferred into 15-mL conical tubes, and subsequently centrifuged at 402× *g* and 4 °C for 5 min (modified from Loftus et al. [27]). The supernatant was aspirated and discarded; the pellet was resuspended in 4 mL ice-cold PBS, and centrifuged at 402× *g* and 4 °C for 5 min. Next, the supernatant was aspirated and discarded, and the upper white layer of the pellet, which contained PMN, was collected and transferred into a 1.5-mL tube. Samples were analyzed with a hematological cytometer (ADVIA 120 Siemens Medical Solutions) to determine the concentration of PMN per mL. Cell concentration was adjusted to 100 × 10^6^ PMN/mL in ice-cold PBS.

### 4.3. Experiments 2 and 3: Effects of SP and Sperm Concentration on PMN-sperm Binding

#### 4.3.1. Semen Collection and Process

Semen was collected with an artificial vagina (Hannover model) equipped with an in-line filter (Minitüb Ibérica SL; Reus, Spain) to get a gel-free semen sample. Semen volume was recorded and diluted 1:1 (*v*:*v*) with a skim-milk-based semen extender [39]. Sperm concentration was evaluated using a Neubauer chamber (Paul Marienfeld GmbH & Co. KG; Lauda-Königshofen, Germany), and four treatments containing different sperm concentrations (100 × 10^6^, 200 × 10^6^, 500 × 10^6^, and 1000 × 10^6^ sperm/mL) were prepared.

#### 4.3.2. Treatments

Eleven treatments containing different sperm concentrations, PMN, FMLP and DMSO were prepared, as follows. In the case of Experiment 2, which tested the role of FMLP activation on PMN [26], these treatments were: (a) PMN + 100 × 10^6^ sperm/mL + SP + FMLP 0.1 mM; (b) PMN+ 200 × 10^6^ sperm/mL + SP + FMLP 0.1 mM; (c) PMN + 500 × 10^6^ sperm/mL + SP+ FMLP 0.1 mM; (d) PMN + 1000 × 10^6^ sperm/mL + SP + FMLP 0.1 mM; (e) PMN + 500 × 10^6^ sperm/mL + SP + DMSO; (f) 500 × 10^6^ sperm/mL + SP + DMSO; and (g) control (500 × 10^6^ sperm/mL + SP sample). Treatments g and f were the controls for DMSO, which was the vehicle for FMLP dilution. Treatment g was the control for sperm viability and motility throughout incubation. In the case of Experiment 3, which aimed at assessing the effects of different sperm concentration on sperm:PMN binding, we tested the following treatments: (a) PMN + 100 × 10^6^ sperm/mL + SP; (b) PMN + 200 × 10^6^ sperm/mL + SP; (c) PMN + 500 × 10^6^ sperm/mL + SP; and (d) PMN + 1000 × 10^6^ sperm/mL + SP.

All semen samples were incubated in a water bath at 37 °C, and their viability, sperm-PMN binding and motility evaluated after 0 h, 1 h, 2 h, 3 h and 4 h of incubation.

#### 4.3.3. Evaluation of Sperm Viability

Sperm viability was determined through eosin-nigrosin staining [40]. A minimum of 200 spermatozoa/sample were examined under a bright-field, optical microscope (Olympus Europe, Hamburg, Germany) at 1000 × magnification. We recorded percentages of viable spermatozoa (eosin-negative), and those of viable spermatozoa attached to PMN.

#### 4.3.4. Sperm–PMN Binding

Sperm–PMN attachment was determined as previously described [20,41]. In brief, 10 µL of each sample was placed onto a slide prior to mixing with Diff-Quick stain (QCA, Amposta, Spain) and smearing [22]. A minimum of 200 sperm cells were counted under a bright-field, optical microscope at 1000 × magnification. The following three parameters were recorded: (a) percentages of spermatozoa attached to PMN; (b) sperm:PMN ratio (number of spermatozoa attached per PMN); and (c) percentages of spermatozoa attached to PMN either via their head or their tail.

#### 4.3.5. Sperm Motility

Sperm motion characteristics in the free (unattached) sperm population were evaluated by means of a computer assisted sperm analysis (CASA) system (Integrated Semen Analysis System, ISAS^®^ Ver.1.0.15; Projects and Services R+D SL, Proiser; Valencia, Spain). This system consisted of a negative phase-contrast microscope (Olympus BH-2) with a yellow light filter and a warm-up plate, and a digital video camera (Basler, Ahrensburg Germany) connected to a computer containing the ISAS software. Samples were placed into a pre-warmed Neubauer chamber (37 °C) and examined at 1000 × magnification, with at least 200 spermatozoa being counted per analysis.

### 4.4. Statistical Analyses

A statistical package (IBM SPSS Statistics 25.0; Armonk, New York, NY, USA) was used to analyze the results obtained in this work. Shapiro-Wilk and Levene tests were conducted to assess data distribution and homogeneity of variances, respectively. When required, data were transformed through arcsine √x. Following this, one-way ANOVA (experiment 1) or a linear mixed model (experiments 2 and 3) followed by post-hoc Sidak test for pair-wise comparisons were run. The treatment was considered to be the fixed-effects factor, the donkey was the random-effects factor and the incubation time was the intra-subjects factor. When, even after linear transformation, data did not show normal distribution and/or variances were not homogenous, non-parametric alternatives (Kruskal-Wallis and Mann-Whitney tests for Experiment 1; and Friedman and Wilcoxon tests for Experiments 2 and 3) were used. The level of significance was set at *p* ≤ 0.05, and data are shown as mean ± standard error of the mean (SEM).

## 5. Conclusions

In conclusion, the whole semen, including seminal plasma, activates PMN and induces sperm-PMN binding in jennies. However, we observed low percentages of phagocytosis upon sperm:PMN binding, and not only were the percentages of viable spermatozoa higher in the bound-to-PMN than in the free sperm population, but sperm cells that were apparently released from PMN after 3 h of incubation also showed good motility. Therefore, the fact that sperm previously attached to PMN could be later released, especially when concentration was at least 500 × 10^6^ spermatozoa/mL, suggests that this binding does not necessarily lead to the induction of phagocytosis. For this reason, and together with the fact that NETosis has been extensively reported on in other mammalian species, further research should be aimed at elucidating: (a) through which ligand/receptor spermatozoa and PMN bind; (b) whether the contact between spermatozoa and PMN in the presence of seminal plasma always triggers phagocytosis or rather some sperm cells remain fertile upon releasing from PMN, and (c) if, in the donkey, clearance of spermatozoa through PMN also occurs via NETosis rather than phagocytosis. In the latter case, while one would expect that this could also be the case, the previously reported variations between other species along with the differences related to the semen deposition site support the need for specific experiments in the donkey.

## Figures and Tables

**Figure 1 ijms-21-03478-f001:**
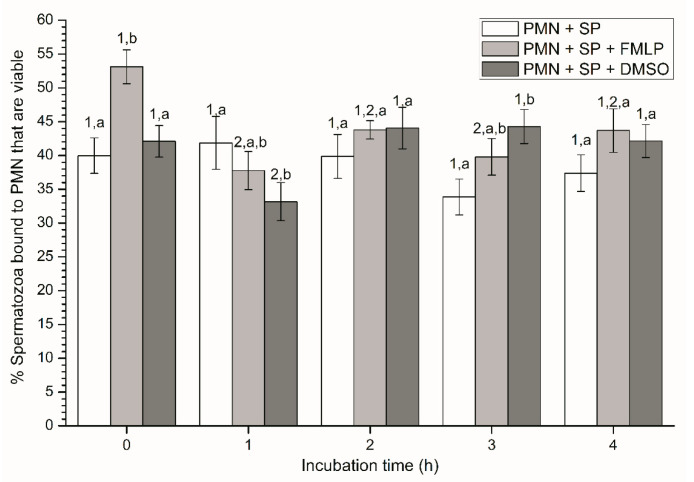
Percentages of viable spermatozoa bound to PMN in the three treatments (PMN + SP, PMN +SP + FMLP and PMN + SP + DMSO) throughout the incubation time (4 h; Experiment 2). Different superscripts (a, b) mean significant (*p* < 0.05) differences between treatments within a given time point, and different numbers (1, 2) mean significant (*p* < 0.05) differences between incubation times within a given treatment. Data are shown as mean ± SEM for seven experiments. Sperm concentration: 500 × 10^6^ sperm/mL. PMN concentration: 100 × 10^6^ PMN/mL. PMN: polymorphonuclear neutrophils; SP: seminal plasma; DMSO: dimethyl sulfoxide; FMLP: formyl-methionyl-leucyl-phenylalanine.

**Figure 2 ijms-21-03478-f002:**
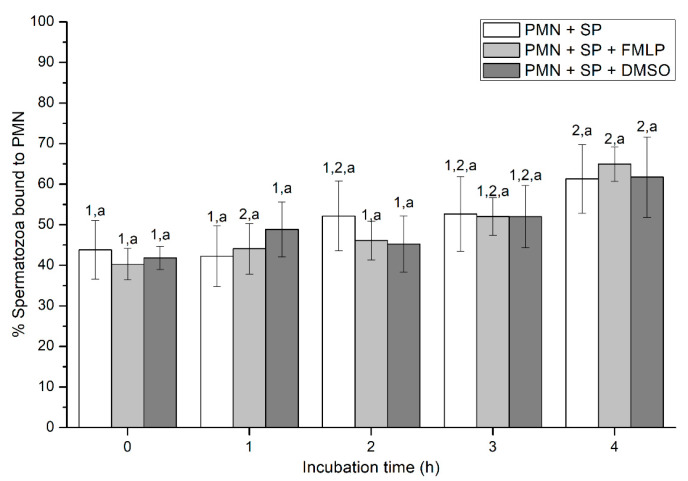
Percentages of spermatozoa bound to PMN in the three treatments (PMN + SP, PMN + SP + FMLP and PMN + SP + DMSO) throughout the incubation time (4 h; Experiment 2). Different superscripts (a, b) mean significant (*p* < 0.05) differences between treatments within a given time point, and different numbers (1, 2) mean significant (*p* < 0.05) differences between incubation times within a given treatment. Data are shown as mean ± SEM for seven experiments. Sperm concentration: 500 × 10^6^ sperm/mL. PMN concentration: 100 × 10^6^ PMN/mL. PMN: polymorphonuclear neutrophils; SP: seminal plasma; DMSO: dimethyl sulfoxide; FMLP: formyl-methionyl-leucyl-phenylalanine.

**Figure 3 ijms-21-03478-f003:**
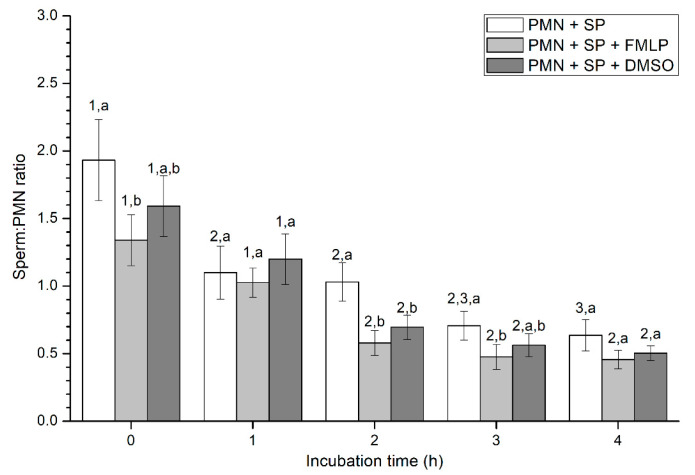
Number of spermatozoa bound to PMN (sperm:PMN ratio) in the three treatments (PMN+SP, PMN+SP+FMLP and PMN+SP+DMSO) throughout the incubation time (4 h; Experiment 2). Different superscripts (a, b) mean significant (*p* < 0.05) differences between treatments within a given time point, and different numbers (1, 2) mean significant (*p* < 0.05) differences between incubation times within a given treatment. Data are shown as mean ± SEM for seven experiments. Sperm concentration: 500 × 10^6^ sperm/mL. PMN concentration: 100 × 10^6^ PMN/mL. PMN: polymorphonuclear neutrophils; SP: seminal plasma; DMSO: dimethyl sulfoxide; FMLP: formyl-methionyl-leucyl-phenylalanine.

**Figure 4 ijms-21-03478-f004:**
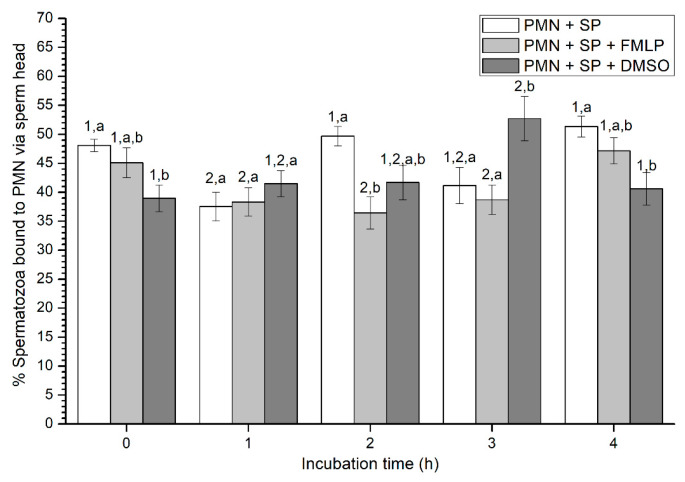
Percentages of viable spermatozoa bound to PMN via sperm head in the three treatments (PMN + SP, PMN + SP + FMLP and PMN + SP + DMSO) throughout the incubation time (4 h; Experiment 2). Different superscripts (a, b) mean significant (*p* < 0.05) differences between treatments within a given time point, and different numbers (1, 2) mean significant (*p* < 0.05) differences between incubation times within a given treatment. Data are shown as mean ± SEM for seven experiments. Sperm concentration: 500 × 10^6^ sperm/mL. PMN concentration: 100 × 10^6^ PMN/mL. PMN: polymorphonuclear neutrophils; SP: seminal plasma; DMSO: dimethyl sulfoxide; FMLP: formyl-methionyl-leucyl-phenylalanine.

**Figure 5 ijms-21-03478-f005:**
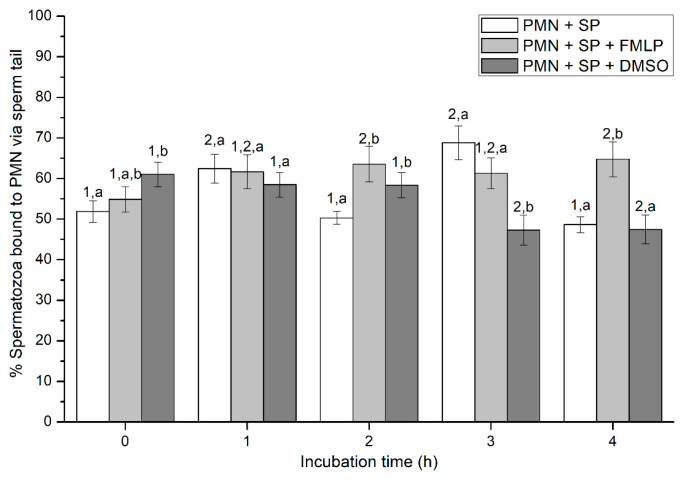
Percentages of viable spermatozoa bound to PMN via sperm tail in the three treatments (PMN + SP, PMN + SP + FMLP and PMN + SP + DMSO) throughout the incubation time (4 h; Experiment 2). Different superscripts (a, b) mean significant (*p* < 0.05) differences between treatments within a given time point, and different numbers (1, 2) mean significant (*p* < 0.05) differences between incubation times within a given treatment. Data are shown as mean ± SEM for seven experiments. Sperm concentration: 500 × 10^6^ sperm/mL. PMN concentration: 100 × 10^6^ PMN/mL. PMN: polymorphonuclear neutrophils; SP: seminal plasma; DMSO: dimethyl sulfoxide; FMLP: formyl-methionyl-leucyl-phenylalanine.

**Figure 6 ijms-21-03478-f006:**
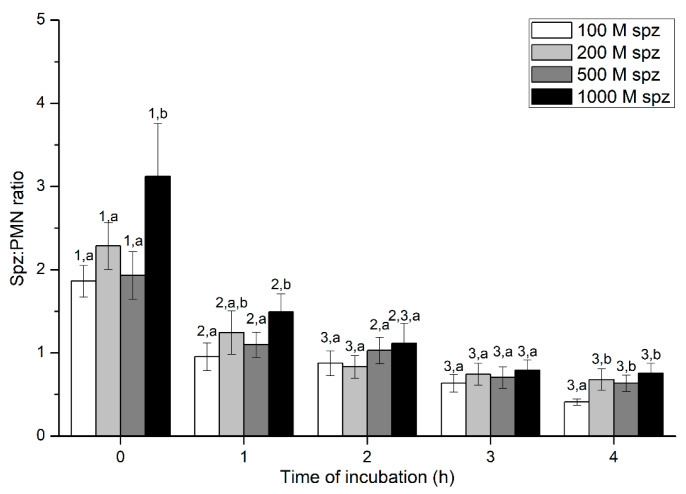
Number of spermatozoa bound to PMN (sperm:PMN ratio) after incubation with four different sperm concentrations (100 × 10^6^, 250 × 10^6^, 500 × 10^6^, and 1000 × 10^6^ spermatozoa/mL) for 4 h (Experiment 3). Different superscripts (a, b) mean significant (*p* < 0.05) differences between treatments within a given time point, and different numbers (1, 2) mean significant (*p* < 0.05) differences between incubation times within a given treatment. Data are shown as mean ± SEM for seven experiments. PMN concentration: 100 × 10^6^ PMN/mL.

**Figure 7 ijms-21-03478-f007:**
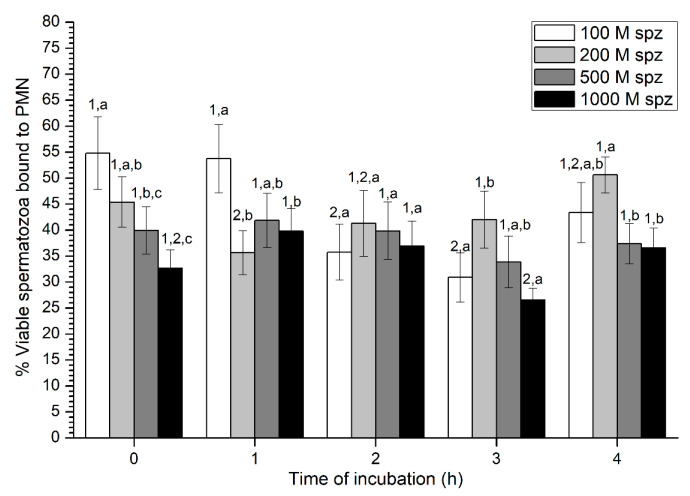
Percentages of viable spermatozoa bound to PMN after incubation with four different sperm concentrations (100 × 10^6^, 250 × 10^6^, 500 × 10^6^, and 1000 × 10^6^ spermatozoa/mL) for 4 h (Experiment 3). Different superscripts (a, b) mean significant (*p* < 0.05) differences between treatments within a given time point, and different numbers (1, 2) mean significant (*p* < 0.05) differences between incubation times within a given treatment. Data are shown as mean ± SEM for seven experiments. PMN concentration: 100 × 10^6^ PMN/mL.

**Table 1 ijms-21-03478-t001:** PMN concentration (×10^3^ cells/µL) obtained after conducting different PMN isolation protocols (Experiment 1).

Protocol	[PMN] (×10^3^ cells/µL)
1	8.23 ± 0.87 ^a^
2	6.20 ± 1.23 ^a,b^
3	3.37 ± 0.95 ^b^
4	56.16 ± 6.31 ^c^
5	93.70 ± 7.07 ^d^
6	126.07 ± 10.01 ^d^
7	307.07 ± 19.77 ^e^

Different superscript letters (a, b, c, d, e) mean significant (*p* < 0.05) differences between protocols. Data are shown as mean ± SEM for seven separate experiments.

**Table 2 ijms-21-03478-t002:** Sperm viability in different treatments and after incubation at 37 °C for 4 h (Experiment 2). Sperm concentration was set at 500 × 10^6^ sperm/mL in all cases.

Treatment	0 h	1 h	2 h	3 h	4 h
PMN + SP	77.9 ± 4.7 ^1, a^	65.2 ± 7.0 ^1, 2, a^	55.8 ± 6.0 ^2, 3, a^	43.2 ± 4.3 ^3, a^	30.5 ± 3.3 ^4, a^
PMN + SP + FMLP	73.9 ± 6.8 ^1, a^	62.5 ± 7.1 ^1, 2, a^	53.6 ± 7.1 ^2, 3, a^	38.7 ± 4.5 ^3, 4, a^	28.9 ± 3.4 ^4, a^
PMN + SP + DMSO	74.9 ± 7.4 ^1, a^	62.3 ± 8.1 ^1, 2, a^	54.7 ± 6.8 ^2, 3, a^	43.0 ± 8.3 ^3, 4, a^	30.7 ± 7.3 ^4, a^
SP + DMSO	74.5 ± 6.1 ^1, a^	64.2 ± 7.0 ^1, 2, a^	55.0 ± 6.8 ^2, 3, a^	43.3 ± 4.8 ^3, a^	31.2 ± 4.9 ^4, a^
SP	77.7 ± 6.3 ^1, a^	68.1 ± 5.3 ^1, 2, a^	55.5 ± 5.8 ^2, 3, a^	45.1 ± 6.5 ^3, a^	28.3 ± 4.8 ^4, a^

Different superscripts (a, b) mean significant (*p* < 0.05) differences between treatments within a given time point, and different numbers (1, 2) mean significant (*p* < 0.05) differences between incubation times within a given treatment. Data are shown as mean ± SEM for seven experiments. PMN: polymorphonuclear neutrophils; SP: seminal plasma; DMSO: dimethyl sulfoxide; FMLP: formyl-methionyl-leucyl-phenylalanine.

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
