# Peer review of "Seminal Plasma, Sperm Concentration, and Sperm-PMN Interaction in the Donkey: An In Vitro Model to Study Endometrial Inflammation at Post-Insemination"

_ijms, 2020, doi:10.3390/ijms21103478_

Round 1

Reviewer 1 Report

The he manuscript is well written, but there are some concerns that need to be addressed.

General comments:

  • The second objective to determine whether the presence of seminal plasma and different sperm concentrations affect the percentage of PMN’s bound to sperm cannot be addressed without a negative control group in the absence of seminal plasma. What is the rationale for excluding spermatozoa as activators of PMNs in donkeys?
  • PMN binding and phagocytosis of spermatozoa result in cell death of the PMNs. Could this have skewed your data, specifically the decline of spermatozoa bound to PMNs over time?
  • The conclusion regarding the formation of NETs is not supported without DNA specific staining. In addition, DNAse in seminal plasma is likely to prevent the formation of NETs, which has been described by Alghamdi et al to result in clusters of immotile sperm.
  • Please confirm that the animal experiments were approved by an Institutional Animal Care and Use Protocol.

Specific comments:

Lines 25-26: Neutrophil Extracellular Traps do not involve binding of sperm to the surface of PMNs. Please reconsider this conclusion without specific DNA staining to confirm the presence of NETs.

Line 59: Please remove one of the repeated “them”.

Lines 80-81: See general comments. This objective cannot be addressed without a seminal plasma free control.

Table 2: I assume you mean Hours rather than Days. Please correct.

Lines 120-123: This paragraph does not appear to be supported by figure 4. Please explain if I have missed something.

  • No significant differences among treatments within time points other than T0. However, SP is labeled in the graph as being significantly different to SP+DMSO at T1.
  • Please explain how SP and SP+DMSO are similar (a), while SP+FMPL is labeled (ab) in spite of data showing this treatment between the other two?
  • No significant differences among treatments along incubation times. However, significant differences are indicated in the graph for SP+FMLP at T0 versus T1 and T3. Significant differences are also indicated in the graph for SP+DMSO at T1 versus all other times.

Line 123: Please number the figures in the order they appear in the manuscript. Figure 4 should not be proceeding figures 2 and 3.

Lines 165-167: Could this be the results of non-observed phagocytosis and cell death between observations?

Line 201: Please remove the word “show” and add “reports” for the sentence to read “These data are in agreement with previous reports in donkeys, in which…….”

Line 208: Could this be explained by the fairly rapid cell death of PMNs soon after phagocytosis?

Lines 212-221: The data does not support this conclusion without DNA staining. Please remove or re-word.

Lines 220-221: This statement is not supported by the data. Please remove.

Line 392: Be careful with the conclusions on NETs without any evidence that this took place.

Author Response

Reviewer #1

Dear Reviewer,

Thank you very much for your comments and suggestions. We have tried our best to address all your concerns, and we have made the changes to the Manuscript accordingly.

General comments

Comment 1
Reviewer: The manuscript is well written, but there are some concerns that need to be addressed.
Answer: Thank you very much for your positive assessment and extensive feedback. Your comments have definitely improved the quality of our Manuscript, and we really appreciate the time you took to review our paper.

Comment 2
Reviewer: The second objective to determine whether the presence of seminal plasma and different sperm concentrations affect the percentage of PMN’s bound to sperm cannot be addressed without a negative control group in the absence of seminal plasma. What is the rationale for excluding spermatozoa as activators of PMNs in donkeys?
Answer: Thank you very much for your comment. We are willing to accept that without the aforementioned negative control (i.e. without seminal plasma, SP), we cannot exclude the possibility that sperm cells activate PMN. In this context, however, we would like to indicate that our main aim was to investigate the effects of sperm concentration (in the presence of SP) since, because we are aware that it is higher in jackasses than in stallions, we deemed this subject to be worth enough to be investigated. In addition, we would also like to point out that we focused on the effects of sperm concentration in the presence of SP because previous works (Miró et al. Anim Reprod Sci 2013, 140: 164-172; Vilés et al. Anim Reprod Sci 2013, 143: 57-63) and other unpublished results from our group suggest that SP plays a crucial role in modulating the interaction between spermatozoa and PMN. All that being said and because it is true that we cannot exclude the role of sperm cells themselves, we have, following your comment, rewritten the second objective, making clear that our study is focused on the whole semen (i.e. sperm + SP) and that further studies should address whether isolated sperm cells play any role with regard to sperm-PMN interaction.

Comment 3
Reviewer: PMN binding and phagocytosis of spermatozoa result in cell death of the PMNs. Could this have skewed your data, specifically the decline of spermatozoa bound to PMNs over time?
Answer: Thank you very much for your comment. While, as you rightly pointed out, phagocytosis results in cell death, the levels of phagocytosis observed herein were very low. Therefore, although we cannot completely exclude that phagocytosis could have skewed our data, the fact that phagocytosis was low leads us to suggest that its impact on the global results was not much apparent.

Comment 4
Reviewer: The conclusion regarding the formation of NETs is not supported without DNA specific staining. In addition, DNAse in seminal plasma is likely to prevent the formation of NETs, which has been described by Alghamdi et al to result in clusters of immotile sperm.
Answer: We understand your comment, and we think that the Manuscript could be misleading here. The aim of this study was not to investigate NETosis, but we rather focused on the role of sperm/SP with regard to sperm:PMN interaction. Therefore, by including such a sentence in the Discussion we just aimed at suggesting that NETs could be formed when sperm and PMN interact. In fact, we are currently developing a new study focusing on NETosis in the donkey and, amongst other tests, we are performing DNA- and elastase-staining. Following your comment, the conclusion has been revised.

Comment 5
Reviewer: Please confirm that the animal experiments were approved by an Institutional Animal Care and Use Protocol.
Answer: Thank you very much for your comment. Experiments were approved by the Ethics Committee, Autonomous University of Barcelona (Bellaterra, Cerdanyola del Vallès). This information has been added to M&M section.

Specific comments:

Comment 6
Reviewer: Lines 25-26: Neutrophil Extracellular Traps do not involve binding of sperm to the surface of PMNs. Please reconsider this conclusion without specific DNA staining to confirm the presence of NETs.
Answer: Sorry for this misleading sentence. This was just a suggestion rather than a conclusion reached from the results obtained in this study. Therefore, we agree with you that we cannot make that assertion without including specific experiments aimed at identifying the occurrence of NETosis. As explained before, we are currently conducting a new work focused on the formation of NETs. In the light of the aforementioned and following your comment, this sentence has been revised.

Comment 7
Reviewer: Line 59: Please remove one of the repeated “them”.
Answer: Thank you very much for drawing our attention into this mistake. This repeated word has been removed.

Comment 8
Reviewer: Lines 80-81: See general comments. This objective cannot be addressed without a seminal plasma free control.
Answer: Thank you very much for your comment. As answered to Comment 2, the second objective has been rewritten making clearer that our work was conducted using the whole semen.

Comment 9
Reviewer: Table 2: I assume you mean Hours rather than Days. Please correct.
Answer: You are entirely right. This mistake has been corrected.

Comment 10
Reviewer: Lines 120-123: This paragraph does not appear to be supported by figure 4. Please explain if I have missed something.
Answer: You are right. While Sp+DMSO at T1 showed a significant decrease in sperm survival, no significant differences in the percentages of spermatozoa bound to PMN were found when T0 and T2, T3 and T4 were compared. In order to clarify this point, and following your comment, we have added that explanation in the text.

Comment 11
Reviewer: No significant differences among treatments within time points other than T0. However, SP is labeled in the graph as being significantly different to SP+DMSO at T1.
Answer: Thank you for your comment. Different superscript letters (a, b) in the Figure mean significant (p<0.05) differences between treatments within a given time point. At 1 h, there were significant differences between PMN/SP and PMN/SP/DMSO, but not between PMN/SP and PMN/SP/FMLP or between PMN/SP/FMLP and PMN/SP/DMSO. That being said, we found a mistake for PMN/SP/FMLP at 3 h in Fig. 4 (only one superscript) that has been corrected.

Comment 12
Reviewer: Please explain how SP and SP+DMSO are similar (a), while SP+FMPL is labeled (ab) in spite of data showing this treatment between the other two?
Answer: There were no significant differences within the SP+FMLP treatment when 0 h was compared to 2 h or to 4 h. However, there was a significant decrease at 1 h. We have revised the text to make this clearer.

Comment 13
Reviewer: No significant differences among treatments along incubation times. However, significant differences are indicated in the graph for SP+FMLP at T0 versus T1 and T3. Significant differences are also indicated in the graph for SP+DMSO at T1 versus all other times.
Answer: We are willing to accept that this paragraph needed to be rewritten to explain our results better. For this reason, and following your comment, this paragraph has been rewritten (LL120-123).

Comment 14
Reviewer: Line 123: Please number the figures in the order they appear in the manuscript. Figure 4 should not be proceeding figures 2 and 3.
Answer: Thank you for your correction. Numbering of Figures in this revised version has been modified. Thus, Fig. 4 has been renumbered to Fig 2, Fig. 2 has been renumbered to Fig. 3, and Fig. 3 has been renumbered to Fig. 4.

Comment 15
Reviewer: Lines 165-167: Could this be the results of non-observed phagocytosis and cell death between observations?
Answer: Yes, it could. Your suggestion has been considered in the Discussion.

Comment 16
Reviewer: Line 201: Please remove the word “show” and add “reports” for the sentence to read “These data are in agreement with previous reports in donkeys, in which…….”
Answer: This has been corrected in the Manuscript.

Comment 17
Reviewer: Line 208: Could this be explained by the fairly rapid cell death of PMNs soon after phagocytosis?
Answer: Your suggestion is very interesting and has been included in the discussion.

Comment 18
Reviewer: Lines 212-221: The data does not support this conclusion without DNA staining. Please remove or re-word.
Answer: Thank you very much for your comment. This paragraph has been revised and rewritten.

Comment 19
Reviewer: Lines 220-221: This statement is not supported by the data. Please remove.
Answer: This sentence has been removed.

Comment 20
Reviewer: Line 392: Be careful with the conclusions on NETs without any evidence that this took place.
Answer: We agree with you. Conclusions have been rewritten taking into account your comment.

Reviewer 2 Report

Dear Authors,

The comments are in a attached file.

Best regards.

Author Response

Reviewer #2

Dear Reviewer,

Thank you very much for your comments and suggestions. We have tried our best to address all your concerns, and we have made the changes to the Manuscript accordingly.

General Comments

Comment 1
Reviewer: The authors meant to investigate the role of seminal plasma and sperm concentration on jenny post-artificial insemination using an in vitro study. This is a very innovative work. For that purpose, the authors retrieved peripheral blood PMN and performed very elegant in vitro studies where they evaluated the interaction of PMN (either unstimulated or stimulated by FMLP) with sperm cells in the presence of seminal plasma. Since PMN were vital for this study, it would be worth to refer PMN in the title. The title refers to jenny post-artificial insemination, which might predispose the reader to expect an in vivo study. Therefore, I would suggest that the title was modified accordingly to the content of the manuscript, in order to include PMN which are involved. The construction of the manuscript is very appealing, and the rationale for the study is adequate and demonstrates the need of an in vitro study to answer queries that are brought up from donkey reproduction clinics. The methodology used is adequate. However, English should be improved. After the authors address the issues raised, I would suggest this manuscript be considered for publication in the International Journal of Molecular Sciences.
Answer: Thank you very much for your kind review and for the time dedicated to provide this extensive feedback. The title has been revised as per your suggestion.

Specific Comments

Comment 2
Reviewer: L20-21 – Please, write this sentence as follows: We then interrogated ourselves on how PMN, which require an activating agent like formyl-methionyl-leucyl-phenylalanine (FMLP) to be activated, are affected by seminal plasma (SP). Finally, we
Answer: Thank you very much for your comment. This sentence has been corrected.

Comment 3
Reviewer: L22- It should read: tested how four different sperm concentrations
Answer: Corrected as requested.

Comment 4
Reviewer: L23- It should read: We observed …
Answer: Corrected as requested.

Comment 5
Reviewer: L28.29 - It should read: …percentage of unattached spermatozoa to PMN
Answer: Corrected as requested.

Comment 6
Reviewer: L30- It should read: …of sperm phagocytosed by PMN is low
Answer: Corrected as requested.

Comment 7
Reviewer: L38- It should read: and milk, as an alternative
Answer: Corrected as requested.

Comment 8
Reviewer: L36.39- Since this is a very long sentence I would suggest to divide it, as follows: However, there has been, in the recent years, an increased global demand for donkey products. As such, donkey skin, which has been used in China to make the traditional donkey-hide gelatin (Ejiao); meat; and milk, as an alternative to bovine milk for infants with some allergic and inflammatory conditions, and also used in the cosmetic industry.
Answer: Thank you very much for your comment. This sentence has been revised according to your suggestion.

Comment 9
Reviewer: L46 It should read: jackass sperm cells have good post-thaw viability and motility [4][5][6], they are able to penetrate in vitro …
Answer: Corrected as requested.

Comment 10
Reviewer: L56- Is this inflammatory response a physiological one as in the mare?
Answer: Yes, as explained in the Manuscript, there is a physiological post-AI inflammatory response in both species (mare and jenny) but this response is significantly higher in the jenny.

Comment 11
Reviewer: L58- Since spermatozoa is plural you should write provoke and not provokes.
Answer: Corrected as requested.

Comment 12
Reviewer: L59. The word “when” is repeated.
Answer: Corrected as requested.

Comment 13
Reviewer: L70 It should read: since it helps to avoid
Answer: Corrected as requested.

Comment 14
Reviewer: L71 It should read: which SP modulates
Answer: Corrected as requested.

Results

Comment 15
Reviewer: L104 Which one is treatment T?
Answer: Corrected as requested.

Comment 16
Reviewer: L115. You mentioned that sperm viability decreased significantly (p<0.05) after 3 h of incubation. What about PMN viability? Was it assessed? Did the spz only attached to viable PMN or also to dead PMN? If you have that data it would be important to share it.
Answer: Thank you for your comment. While we agree with you that including PMN viability could help us understand sperm:PMN interaction, we do not unfortunately have those data. That being said, we will incorporate the evaluation of this parameter in our further studies. Thank you very much for your suggestion.

Comment 16
Reviewer: L138- “differnt” is misspelled. Please, correct it.
Answer: Corrected as requested.

Comment 17
Reviewer: L147- “Percentage” instead of percentages.
Answer: Corrected as requested.

Comment 18
Reviewer: In fig. 1 you show the percentage of Spz bound to PMN at different incubation times, with only SP, or adding FMLP or DMSO. In Fig. 2 you show the Average number of Spz bound to one PMN at different incubation times. However, it is not clear which spz and PMN concentrations were used. Was 500x106 sperm/mL used? If that is the case this should be clearly stated, also in the Figure legends.
Answer: Thank you very much for your comment. Sperm and PMN concentration in treatments drawn in each Figure have been mentioned.

Comment 19
Reviewer: L143- In figure 4 legend you should mention that you refer to viable spermatozoa.
Answer: Corrected as requested.

Comment 20
Reviewer: L163- You said that spermatozoa/mL on sperm-PMN binding only took into consideration those treatments that did not contain either FMLP or DMSO. Do you mean that data on spermatozoa/mL on sperm-PMN binding were only considerate in those treatments that did not include either FMLP or DMSO exposure (Treatments E, F, G, H).? According to your experimental design, E,F,G and H treatments included FMLP. Please, clarify this.
Answer: Thank you for your comment and suggestions. In order to address this comment, we have revised the experimental design section and clarified these points.

Comment 21
Reviewer: L188- please use uterine flushing instead of uterine washings.
Answer: Corrected as requested.

Comment 22
Reviewer: L191 – “by isolating” instead of “through isolating”
Answer: Corrected as requested.

Comment 23
Reviewer: L199- Please write it, as follows: ….not alter the percentage of spermatozoa bound to PMN, or their viability either free or attached to PMN populations, along the incubation time.
Answer: Thank you very much for your comment. The sentence has been rewritten following your request.

Comment 24
Reviewer: L201- Please write it, as follows: “These data are in agreement with previous one ”
Answer: Corrected as requested.

Comment 25
Reviewer: L232-233- In addition, sperm:PMN ratios also decreased throughout incubation time and the percentage sof viable but the percentages of spermatozoa attached PMN increased throughout the successive incubation times. This sentence os not very clear. Do you mean the percentage of viable spz or viable PMN?
Answer: Thank you very much for your comment. We have revised and rewritten this sentence accordingly.

Comment 26
Reviewer: L236 – You say that “This means that high Spz concentrations have higher possibilities to have free Spz at 3-4h”. Do you mean that higher spz concentration will not increase the number of spz bound to PMN, and therefore more spz will be free in the medium at 3-4h?
Answer: Thank you for your comment. As explained in the text, sperm motility analysis with CASA system showed that free spermatozoa were immotile at 3-4 h. However, we also observed that when sperm concentration was >500 x106 spermatozoa/mL, some sperm cells became free, as it can be observed in a movie included as Supplementary File 1.

Comment 28
Reviewer: L244- Please correct the verb: An important question is if the spermatozoon that fertilizes is in the population of spermatozoa that become free.
Answer: Corrected as requested.

Comment 29
Reviewer: L 246 -I would suggest this sentence to be written, as follows: Further studies are necessary to analyze the post-breeding uterine PMN-Spz relationship in donkeys, and the effect of seminal plasma in order to control their role in donkey reproductive strategy.
Answer: This sentence has been rewritten as requested.

Materials and Methods

Comment 30
Reviewer: L251- Please write it, as follows: All animals were in good health conditions and of proven fertility. While conducting the experiment, they were housed at the experimental farm, Faculty of Veterinary Science, Autonomous University of Barcelona (Bellaterra, Cerdanyola del Vallès; Spain).
Answer: Corrected as requested.

Comment 31
Reviewer: L290 - two volumes of distilled water
Answer: Corrected as requested.

Comment 32
Reviewer: L360 - The neutrophil fraction
Answer: Corrected as requested.

Comment 33
Reviewer: L318 - and the resulting sample
Answer: Corrected as requested.

Comment 34
Reviewer: L351- …. DMSO were prepared, as follows:
Answer: Corrected as requested.

Comment 35
Reviewer: L350-356- Experiment 2 - I do understand the need for all the groups used. However, it would be easier for the reader if these treatment groups were divided into separate experiments. I would advise the authors to separate them in two experiments, as follows: (Exp. 2.1) to assess the effects of different sperm concentrations, including the experimental groups from (a) to (d); and (Exp. 2.2) to test FMLP activation effect on PMN, from (e) to (k). You should make sure to include te control groups in both experiments.
Answer: Thank you for your comment. The Experimental design section has been revised to make clear that two separate experiments, with different treatments, were conducted.

Comment 36
Reviewer: In the experimental design is not indicated what the number of PMN/mL were used, as well as the amount of SP. This should be indicated here and also in the legend of the figures.
Answer: Corrected as requested.

Comment 37
Reviewer: L358- 359 – When you mention that Treatments E to H had the same sperm concentration, but contained FMLP, it should be explained that sperm concentrations were the same ones used in Exp. 2.1. It might lead the reader to think that in Exp. 2.2 a single sperm concentration was used.
Answer Again, thank you very much for pointing this out to us. As aforementioned, we have revised the Experimental Design section and made clear that two separate experiments were conducted.

Comment 38
Reviewer: L353- please, write the numbers according to the English/American style: 1,000 instead of 1000, throughout the text.
Answer: Corrected as requested.

Comment 39
Reviewer: L360- controls for DMSO, which is needed for FMLP dilution.
Answer: Corrected as requested.

Round 2

Reviewer 1 Report

While the authors have considered and satisfactory addressed the reviewers concerns, there is obviously remaining disagreement between the authors and this reviewer with regards to speculations on NET's and low degree of phagocytosis that are not supported by the data.

Two mechanisms have been suggested for binding of spermatozoa to PMN. (1) a not fully understood ligand/receptor mechanism that is believed to result in phagocytosis, and (2) NET's. While membrane binding through a ligand/receptor mechanism can be visualized under the microscope as direct contact between sperm on the surface of PMNs, NET's have been shown by Alghamdi et al to "trap" spermatozoa in the vicinity of the PMNs (without contact), resulting in a significant loss of sperm motility. ET's are significantly reduced by the presence of seminal plasma, because of the presence of DNAse in this fluid. It is therefore, not convincing that motile sperm bound to PMNs in the presence of seminal plasma, represents NET's as suggested by the authors. Although the manuscript has been somewhat modified to de-emphasize this conclusion, I still believe there is a discrepancy between data and conclusions.

The authors suggest that there was a "low" degree of phagocytosis, but do not provide strong evidence that this was the case, since PMNs die very soon after engulfing spermatozoa and can therefore not always be observed if not monitored constantly.

Since I believe my concerns may represent a difference of opinion between the authors and the reviewer, I leave it to the editor to make a decision if the data from this study support the conclusions

Author Response

Reviewer #1

General Comments

Comment 1: “While the authors have considered and satisfactory addressed the reviewers concerns, there is obviously remaining disagreement between the authors and this reviewer with regards to speculations on NET's and low degree of phagocytosis that are not supported by the data.”
Answer: Thank you very much for your positive assessment, and for indicating that we considered the previous reviewers’ feedback. In the preceding revision round, we already tried to make the point on the fact that formation of NETs in response to the exposure of sperm to PMN was just a suggestion, and that the aim of the current study was not to evaluate whether NETosis is induced when sperm are co-incubated with PMN. Incidentally, we are now conducting a separate work focused on that issue.

Comment 2: “Two mechanisms have been suggested for binding of spermatozoa to PMN. (1) a not fully understood ligand/receptor mechanism that is believed to result in phagocytosis, and (2) NET's. While membrane binding through a ligand/receptor mechanism can be visualized under the microscope as direct contact between sperm on the surface of PMNs, NET's have been shown by Alghamdi et al to "trap" spermatozoa in the vicinity of the PMNs (without contact), resulting in a significant loss of sperm motility. ET's are significantly reduced by the presence of seminal plasma, because of the presence of DNAse in this fluid. It is therefore, not convincing that motile sperm bound to PMNs in the presence of seminal plasma, represents NET's as suggested by the authors. Although the manuscript has been somewhat modified to de-emphasize this conclusion, I still believe there is a discrepancy between data and conclusions.”
Answer: We completely agree with the reviewer, and we apologize for not having made that agreement clear in the Manuscript.

With regard to the percentages of sperm phagocytosis, we agree with the reviewer that further research should address, when jackass sperm are incubated with jenny PMN, through which ligand/receptor mechanism this phenomenon occurs. This has been mentioned in the Manuscript. Related with this, we would like to indicate that, given the relevance in the field and because the Reviewer #1 explicitly mentioned it in their report, we again read the work by Alghamdi et al. (2009; Anim Reprod Sci 114: 331-344) with great interest. Unfortunately, we could not find how cells were stained when sperm:PMN binding was evaluated under a bright-field microscope. It looks like these authors determined the percentages of spermatozoa bound to PMN through dividing the number of free spermatozoa after 180 min of incubation by those observed at 0 min. Alghamdi et al. (2009) also conducted nice SEM observations, but those analyses were basically focused on NETs and they could not obviously be quantitative. We also looked at another Manuscript from the same group (Alghamdi et al., 2010; Anim. Reprod. Sci. 121: 249-258) and we appreciated their findings on DNAse activity in the seminal plasma of bulls and stallions. We were happy to see that stallion SP, in line with the fact that this species deposits the semen within the uterus, was found to reduce sperm-PMN binding, which matches with our data and are also in agreement with previous reports conducted in our group (Miró et al., 2013; Anim. Reprod. Sci. 140: 164-172).

As far as NETs are concerned, we have revised our suggestion on the potential role of NETosis, making clearer that, in other species, NETs trap sperm in the vicinity of the PMNs (without contact), which results in a significant loss of sperm motility (Alghamdi et al., 2005; Biol. Rep. 74: 1174–1181). Therefore, in the revised version of this Manuscript, our suggestion about the formation of NETs has not been based on sperm motility data.

On the other hand, according to Alghmadi and Foster (2005), the presence of seminal plasma in the horse decreases the formation of NETs when PMNs are activated by sperm, but not when they are activated by bacteria. In our preliminary data from donkeys (which come from a study we are currently conducting), however, formation of NETs (evaluated with SYTOX and DAPI) seems to be activated by seminal plasma rather than by spermatozoa (see the Figure below).

These surprising data, which appear to be in agreement with that found in cattle (Alghamdi et al., 2009; Fichtner et al. 2020; Biol. Reprod. 102: 902-914), differ from studies conducted in horses (Alghmadi and Foster, 2005). In addition, we also observed and reported in an Abstract accepted for the next SSR conference that, in the donkey, exposure to fresh semen induces higher NETosis than incubation with frozen-thawed sperm (78 ± 5.7% vs. 23 ± 2.4%, respectively; P≤0.01; See Appendix 1), which agrees with that reported by Fichtner et al. (2020) in cattle. While we cannot ascertain the extent of the differences between horses and donkey, which are close phylogenetically species, it is clear that rather than a difference of opinion between the reviewer and the authors, these two species may behave differently and the subject definitely warrants further research. In fact, in a recent review from Schjenken and Robertson (Physiol Rev 2020; 100: 1077-1117), seminal plasma in the donkey was suggested to induce endometrial cytokine expression, regulate sperm interactions with PMN , stimulate NET formation and improve fertility.

Finally, we would like to mention that whether PMN trigger phagocytosis or NETosis depends on particle size, so that while the contact of bacteria with PMN induces phagocytosis, that of higher pathogenic cells (such as the yeast ones) induces NETosis (Branzk et al. 2014; Nat. Immunol. 15: 1017-1025). In addition, it seems that if phagocytosis is inhibited, NETosis is activated. Indeed, as indicated by Branzk et al. (2014), a deficiency in the activity of phagocytic receptors seems to disrupt the size-dependent selectivity of NET release due to inefficient phagocytosis. Blocking dectin-1, a phagocytic receptor present in human PMN, with a specific antibody leads to lower rates of phagocytosis and higher NETosis response. Therefore, we suggest that seminal plasma factors could inhibit PMN receptors involved in phagocytosis induction (such as Dectin-1, as reported by Brzank et al. 2014), which would lead to the induction of NETosis.

Comment 3: “The authors suggest that there was a "low" degree of phagocytosis, but do not provide strong evidence that this was the case, since PMNs die very soon after engulfing spermatozoa and can therefore not always be observed if not monitored constantly.”
Answer: Percentages of sperm phagocytosis by PMN were evaluated as in Miro et al. (2013), and they were low. Those percentages were checked every 60 min so, while it is true that monitoring sperm phagocytosis at shorter time periods could provide some additional information, we do believe that these percentages of phagocytosis were evaluated quite periodically. That being said, we have tone down this statement, indicting that further studies using time-lapse (which was not available when the current study was conducted) are much warranted.

Comment 4: “Since I believe my concerns may represent a difference of opinion between the authors and the reviewer, I leave it to the editor to make a decision if the data from this study support the conclusions”
Answer: We do not honestly think that there are differences of opinion between the authors and the reviewer, but it could be that we were not clear enough in the previous version of this Manuscript. That being said, we have made a further effort to rewrite some passages, and we hope that the reviewer will be happy from the angle that the subject and revision have been tackled.

Appendix: Abstract submitted and accepted in the 53rd Annual Meeting of the SSR

Fresh, but not frozen-thawed, semen induces NETosis in jenny polymorphonuclear cells in a concentration and time dependent manner
Y Mateo-Otero1, F Zambrano2, J Catalán3, M Yeste1, J Miro3* and B Fernandez-Fuertes1
1 Department of Biology, University of Girona, Spain; 2 Department of Preclinical Science, Universidad de La Frontera, Chile; 3 Equine Reproduction Service, Universitat Autònoma Barcelona, Spain

In several species, acute endometritis driven by the recruitment of polymorphonuclear cells (PMNs) occurs in response to semen. Release of DNA from PMNs to form neutrophil extracellular traps (NETs) is stimulated by bull, stallion and human sperm, leading to their entrapment. In mares, this endometrial inflammatory response is more dramatic when exposed to frozen-thawed semen, in comparison to fresh semen. While there is no such evidence of this phenomenon occurring in jenny donkeys, artificial insemination (AI) with frozen semen leads to very poor pregnancy rates. Based on these data, we hypothesised that: 1) NETosis in response to semen also occurs in donkeys; and 2) frozen-thawed semen induces more NETosis than fresh semen in this species. In Experiment 1, PMNs from jennies (n=4) were isolated by centrifugation of whole blood through a density gradient. After confirming the presence of >90% PMNs by flow cytometry, cells were incubated in the presence or absence (control) of fresh sperm (1:1, 1:2 or 1:5 PMN:sperm ratios) from one jackass. After 2h or 4h incubation, cells were fixed, stained with Sytox, and the percentage of PMNs that underwent NETosis was determined. Although NETosis increased in all groups from 2 to 4h, more PMNs had reacted in the 1:2 and 1:5 groups, but not in the 1:1, compared to the control (2h control: 27 ± 8.6% vs. 1:2: 47 ± 8.7% and 1:5: 59 ± 8.4%; P≤0.05; 4h control: 50 ± 4.4% vs. 1:2: 89 ± 3.6% and 1:5: 92 ± 3.6%; P≤0.05). No differences were observed in the percentage of reacted PMNs between the 1:2 and 1:5 groups. According to these results, in Experiment 2, PMNs (n=3 jennies) were incubated for 2h in the presence or absence (control) of 1:5 fresh or frozen-thawed semen from the same jackass (n=3). Surprisingly, exposure to fresh semen induced higher NETosis than incubation with frozen-thawed semen (78 ± 5.7% vs. 23 ± 2.4%, respectively; P≤0.01). In addition, no differences were observed between the frozen-thawed group in comparison to the control (23 ± 2.4% vs. 31 ± 3.7%, respectively; P>0.05). In conclusion: 1) both incubation time and fresh sperm concentration positively correlate with the percentage of jenny PMNs that release NETs, however, 2) frozen-thawed semen does not elicit this response. Because samples were prepared following the steps normally used to produce commercial fresh and frozen semen doses, seminal plasma was more diluted in the frozen-thawed samples in comparison to the fresh samples. In addition, a higher percentage of motile sperm were observed in the fresh than in the frozen-thawed semen samples (as one would expect). Thus, these two factors could explained the differences in NETosis reported in the present study. Future experiments will address these observations in order to elucidate the role that NETs play in donkey reproductive physiology.
This work was supported by EU Horizon 2020 Marie Skłodowska-Curie (No 792212).

Round 3

Reviewer 1 Report

The authors have addressed remaining concerns and the manuscript should now be acceptable for publication.